# Thin-Copper-Layer-Induced Early Fracture in Graphene-Nanosheets (GNSs)-Reinforced Copper-Matrix-Laminated Composites

**DOI:** 10.3390/ma15217677

**Published:** 2022-11-01

**Authors:** Hailong Shi, Xiaojun Wang, Xuejian Li, Xiaoshi Hu, Weimin Gan, Chao Xu, Guochao Wang

**Affiliations:** 1State Key Laboratory of Advanced Welding and Joining, Harbin Institute of Technology, Harbin 150001, China; 2GEMS at MLZ, Helmholtz-Zentrum Hereon Lichtenbergstrsasse 1, D-85748 Garching, Germany; 3Center for Analysis and Measurement, Harbin Institute of Technology, Harbin 150001, China; 438th Research Institute, China Electronics Technology Group Corporation, Hefei 230000, China

**Keywords:** graphene nanosheets, laminated composites, fracture mechanism, copper matrix composites, electrodeposition

## Abstract

The strength–ductility trade-off has been a long-standing challenge when designing and fabricating a novel metal matrix composite. In this study, graphene-nanosheets (GNSs)-reinforced copper (Cu)-matrix-laminated composites were fabricated through two methods, i.e., the alternating electrodeposition technique followed by spark plasma sintering (SPS) and direct electrodeposition followed by hot-press sintering. As a result, a Cu-GNS-Cu layered structure formed in the composites with various Cu layer thicknesses. Compared with the pure Cu, the yield strength of the GNS/Cu composites increased. However, the mechanical performance of the GNS/Cu composites was strongly Cu-layer-thickness-dependent, and the GNS/Cu composite possessed a brittle fracture mode when the Cu layer was thin (≤10 μm). The fracture mechanism of the GNS/Cu composites was thoroughly investigated and the results showed that the premature failure of the GNS/Cu composites with a thin Cu layer may be due to the lack of Cu matrix, which can relax the excessive stress intensity triggered by GNSs and delay the crack connection between neighboring GNS layers. This study highlights the soft Cu matrix in balancing the strength and ductility of the GNS/Cu-laminated composites and provides new technical and theoretical support for the preparation and optimization of other laminated metal matrix composites.

## 1. Introduction

It is an efficient method to improve the mechanical performance of the monolithic metal by fabricating metal matrix composites. In general, the component of the composites is carefully selected so that the as-fabricated composites can combine the advantages of both components and exhibit superior mechanical or physical properties than the matrix metal [1,2,3,4]. Thus, during recent decades, designing and adopting new and suitable reinforcements for the metal matrix composites have been of great interest to material researchers. Normally, the reinforcements possess outstanding mechanical or physical properties so that a small-volume addition can bring about great mechanical or physical performance improvement. Recently, the discovery of graphene has provided us with a new member to the reinforcement of metal matrix composites [5]. As a 2D material with a monolayer of SP^2^ bonding carbon, graphene has the highest known intrinsic strength of 130 GPa, a Young’s modulus of 1 TPa, and a high electron mobility of ~15,000 cm^2^ V^−1^ S^−1^, and it has become a popular nanofiller material for polymer matrix composites and metal matrix composites [6,7,8]. Normally, high-quality graphene can be produced by a few methods, including micromechanical exfoliation [9], liquid-phase exfoliation (LPE), chemical vapor deposition (CVD), and element intercalation [10]. However, due to the difficult preparation process and expensive price, graphene nanosheets (GNSs) are usually employed as alternative reinforcements for graphene, which have comparable mechanical properties [11].

Commercially pure copper, while not being the most popular metal used in the industry, has many advantages such as good ductility and chemical stability to experimentally investigate and theoretically validate the basic mechanisms of hardening, softening, or recrystallization in metals [12,13]. Most importantly, there is a good chemical compatibility between graphene and copper even at ultra-high temperature (higher than 1083 °C). This makes it possible for us to fabricate GNS/Cu composites with methods that consist of high-temperature processing procedures [14,15].

To manufacture the GNS/Cu composites, as with other carbonaceous reinforcements—(carbon nanotube, graphene, etc.)—reinforced metal matrix composites, the homogeneous dispersion of the reinforcements and formation of strong bonding between nano-reinforcements and the metal matrix are always considered as the main issues. However, the trade-off between strength and ductility as well as the difficulty of nano-reinforcements incorporation still prevent researchers from successfully manufacturing carbonaceous-reinforcement-reinforced metal matrix composites. Recently, though the architectural design to construct a laminated or gradient structure in the metal matrix composites has been considered to be a promising way to mitigate the conflict of strength and ductility of metal matrix composites [16,17,18,19,20,21,22], several strengthening mechanisms that include the delayed necking [19,20], strain partitioning [18,19], interface constraint [20], and enhanced work hardening rate [21,22], etc., have been proposed and are believed to play a role in strengthening the as-prepared materials. Recently, accumulative roll bonding (ARB), selective coating, electrophoretic deposition (EPD), etc., have been used to fabricate metal-CNTs-laminated composites that realize the high strengthening efficiency of CNTs and good coordination between strength and ductility of the composites [23,24,25]. However, the fabrication process for constructing the laminated structure in the composites is normally too complicated. For instance, to prepare the laminated composites with the ARB process, the as-fabricated composites often need to proceed with cold-rolling for many cycles to adjust the microstructure of the composites [23]. Moreover, the layer thickness of the matrix metal is strongly limited by the initial thickness and the uniform dispersion of the carbonaceous reinforcements cannot be guaranteed. Thus, a novel method that can easily synthesize carbonaceous-reinforcement-reinforced laminated composites with a controllable layer thickness is desired. The EPD process is a simple, scalable, and economical electrochemical process, which has been employed to prepare CNTs/Cu composites with strong interface bonding and a uniform CNT distribution in the composites [26]. GNSs exhibit similar chemical properties to CNTs, and it is reasonable to employ the EPD technique for the preparation of GNS/Cu composites.

In this study, the EPD method was employed to fabricate GNS/Cu-laminated composites in which alternating depositions of GNS and Cu particles were achieved by the EPD technique and the thickness of the Cu layer (less than 5 μm in this work) could be controlled easily by adjusting the deposition time of Cu, which left many choices for subsequent processing. In order to investigate the effects of Cu layer thickness on the deformation and fracture behavior of GNS/Cu composites, GNS/Cu composites with a higher Cu layer thickness (10 μm, 30 μm, and 50 μm) were also fabricated by direct electrodeposition and subsequent hot-press sintering. More importantly, the fracture behavior of the laminated GNS/Cu composites with different Cu layer thicknesses was investigated thoroughly. The role of the soft Cu matrix was highlighted in toughening the laminated composites. This research provides a novel method for the fabrication of laminated-carbonaceous-reinforcement-reinforced metal matrix composites. The important role of Cu matrix in toughening the laminated GNS/Cu composites was revealed, which can be used to optimize other laminated composites with similar structures.

## 2. Experimental Procedures

### 2.1. Raw Materials

The GNSs were purchased from Xiamen Knano Graphene Technology Co., Ltd. (Xiamen, China) produced by the thermal reduction of graphite oxide. The raw GNSs were agglomerated clusters with a fluffy structure and exhibited an irregular-shaped flake morphology with the mean diameter ranging from 1 to ~3 μm. The thickness of the GNSs was around 1~5 nm, which corresponds to less than 15 layers of graphene (assuming that the thickness of a monolayered graphene is 0.34 nm). For the GNS/Cu composites with Cu layer thicknesses of 10 μm, 30 μm, and 50 μm, industrially pure cold-rolled Cu foils with thicknesses of 10 μm, 30 μm, and 50 μm were used. The chemical composition of the Cu foils is displayed in Table 1.

### 2.2. Material Fabrication and Sample Processing

Figure 1 shows the fabrication schematic diagram of the GNS/Cu laminated composites. For the GNS/Cu composites with a Cu layer thickness less than 5 μm, the Cu crystals and the GNSs were alternately deposited on a cathode in two different electrolytic cells. A square Cu plate with a thickness of 1 mm and length of 50 mm was employed as the anode, and the same-sized cold-rolled Cu foil with a thickness of about 10 μm was used as the cathode. The distance between the two electrodes was set to be 80 mm. For the deposition of the GNSs, the electrolyte was prepared by adding 0.04 g of GNSs and 0.06 g of Al(NO)_3_ into 800 mL of isopropanol, and then the mixture was ultrasonically treated for 12 h to obtain a stable GNS liquid suspension. The deposition working voltage was 30 V and the deposition time was 30 s. For the deposition of Cu, saturated copper sulfate solution was employed as the electrolyte. The working voltage was 5 V and the deposition time was 60 s. After the deposition of the Cu crystals and the GNSs for 10 cycles, the as-deposited composite powder was removed from the cathode. For reference, pure Cu powder was prepared using the same process without GNS deposition.

The as-prepared pure Cu powder and the GNS/Cu composite powder were sintered by spark plasma sintering (SPS, SPS-1050) at 950 °C for 10 min under vacuum with a heating rate of 75 °C/min and a sintering pressure of 40 MPa. The size of the obtained cylindrical GNS/Cu composite was 30 mm in diameter and 1.2 mm in thickness. To make the composite denser, a three-pass hot-rolling was conducted at 450 °C to a total thickness reduction of about 50%. Between the passes, the composite samples were isothermally held at 450 °C for 5 min to recover the temperature drop. For comparison, the as-sintered pure Cu samples were also hot-rolled under the same condition.

For the GNS/Cu composites with Cu layer thicknesses of 10 μm, 30 μm, and 50 μm, the cold-rolled Cu foils with different thicknesses (10 μm, 30 μm, and 50 μm) were used and the GNSs were deposited on these Cu foils with the same EPD process. Then, these Cu foils with GNSs on the surface were stacked and sintered by hot-press sintering to prepare the as-sintered GNS/Cu composites. After the sintering process, all the as-sintered GNS/Cu composite samples with different Cu layer thicknesses were proceeded with the hot-rolling process. In detail, each sample was preheated at 450 °C in a resistance furnace for 10 min before rolling. The hot-rolling was conducted for 3 passes for each sample, which reached thickness reductions of 30%, 10%, and 10% with respect to the initial sample thickness. Finally, the hot-rolled samples with a total thickness reduction of around 50% were cooled in air.

### 2.3. Microstructure Characterization

Microstructural features were examined by scanning electron microscopy–electron backscatter diffraction (SEM–EBSD), using a scanning electron microscope (SEM) (Quanta 200FEG) equipped with an EBSD acquisition camera and the Aztech online acquisition software package (Oxford Instruments). For EBSD measurement, the sample was first mechanically polished with 1000-, 2000-, and 4000-grit silicon carbide wet sandpaper, and then electro-polished with a solution of 20% nitric acid and 80% methanol at a temperature lower than 15 °C under a voltage of 10 V for a duration time of 5 s. The SEM–secondary electron (SEM–SE) images were recorded at an accelerating voltage of 20 kV and the EBSD maps were acquired at the same accelerating voltage under the beam-controlled mode with step sizes of 0.2 and 0.1 μm. Energy-dispersive X-ray spectroscopy was used to analyze the chemical composition distribution. The tensile tests were conducted in accordance with **ASTM: E8/E8M-13a** standards. The tensile data (yield strength/YS, ultimate tensile strength/UTS, and elongation/δ) were collected and averaged over 3–5 tests.

## 3. Results

### 3.1. Microstructure of the GNS/Cu Composite Powder and the Cu Foils Deposited with GNSs

Figure 2a displays the SEM–SE image of the raw Cu foil surface that was used as the cathode in the present work. Because the as-received raw Cu foil was in the cold-rolled state, we can see clearly the rolling traces presenting the rolling direction (RD) (as indicated by the yellow arrow). During the EPD process of Cu, Cu nuclei nucleated on the cathode and grew into small Cu particles (as shown in Figure 2b). Notably, the size of the formed Cu powder was controllable by adjusting the deposition time. After the deposition of Cu, the GNSs were deposited on Cu crystals uniformly, as seen in Figure 2c. Due to the 2D geometry of the GNSs, all the GNSs tiled on the Cu particles. In addition, the GNSs possessed a high transparency to electrons through which we can see clearly the Cu particles underneath. This also indicates that the as-deposited GNSs dispersed homogeneously and individually in the suspension after the 12 h ultrasonic treatment and no stacking of the GNSs occurred during the EPD process. Thus, Cu particles and the GNSs were alternatively deposited to form a laminated GNS/Cu composite powder on the Cu cathode. Because the as-deposited GNSs were not fully covered by the deposited Cu particles, we can still see some exposed GNSs in the gap of Cu particles (as circled in Figure 2d).

For the fabrication of the GNS/Cu composites, the GNSs were deposited on Cu foils and the basic unit was prepared as shown in Figure 2e,f. Individual GNSs were observed to distribute on the cold-rolled Cu foils and GNSs possessed similar coverage on the Cu foils with different thicknesses.

### 3.2. Microstructure of Pure Cu and GNS/Cu Composites

Figure 3 displays the EBSD *X*-axis inverse pole figure (IPF-X) images and the {111} pole figures of the hot-rolled pure Cu and the GNS/Cu composites. As can be seen from Figure 3a, the hot-rolled pure Cu still possessed randomly orientated grains with equiaxed grain shape. The individual random orientations of pure Cu can also be inferred from the {111} pole figure in Figure 3c. The irregular {111} pole figure possessed several random distributed strong points, indicating that the grains in pure Cu were randomly orientated and no texture formed after rolling (the maximum orientation intensity value was 4.98). Comparatively, for the GNS/Cu composites, as shown in Figure 3d, a sharper texture formed in the composites and the maximum orientation intensity value was 8.68. As mentioned in the experimental part, the hot-rolling temperature was 450 °C in this work and the total thickness reduction was around 50%, so the obtained texture may not be as sharp as the cold-rolled Cu, due to the orientation randomization caused by the dynamic recrystallization of the deformed Cu. However, because the GNSs tiled on the surface of the Cu powder, they became the Cu layer after the SPS process. In other words, these GNSs were located parallel to the rolling direction. During the hot-rolling process, because of the constraint effect of the GNS layer, a sharper texture could be obtained in the GNS/Cu composite.

The recrystallization process is the reverse process that tends to make the material obtain random grain orientations, specifically, the process that is realized through nucleation of new strain-free grains and grain growth of the new grains. The existence of the GNSs may provide more nucleation sites for the new recrystallized grains. However, the GNSs between different Cu layers can inhibit the migration of the high-angle grain boundaries that prevent the grains from growing too large. Therefore, although the rolling temperature was high, a much smaller grain size was obtained and a comparatively sharper texture formed in the GNS/Cu composites with the maximum orientation intensity value of 8.68 in the {111} pole figure (as displayed in Figure 3d). In addition, the average grain size of the GNS/Cu composite decreased compared with the pure Cu, i.e., the average grain size for the pure Cu was 24.4 ± 0.6 μm and the average grain size for the GNS/Cu composite was 4.1 ± 0.4 μm (as displayed in Figure 3e). During the rolling process, the plastic deformation and the recrystallization process tended to decrease the grain size. However, the high rolling temperature offered a great kinetic driving force for the new crystal nuclei to grow. These two processes conflicted with each other and determined the final grain size of the materials. By contrast, in the GNS/Cu composite, the GNSs between different Cu layers could inhibit the migration of the high-angle grain boundaries, which prevented the grains from growing. As a consequence, the grain size of the GNS/Cu composites decreased a lot in comparison with the pure Cu.

As shown in Figure 3f, in comparison with the pure Cu, the GNS/Cu composite possessed a lower fraction of high misorientation angles, especially a misorientation angle of 60 degrees, which corresponds to the 60°/<111> twin boundary. This may be attributed to the decreased grain size [27]. Interestingly, the GNS/Cu composite had a greater fraction of small-angle misorientations but a lower fraction of high-angle misorientations compared with the pure Cu. In this work, the twins were all growth twins for both the pure Cu and the GNS/Cu composite. For the formation of this kind of twin, the growth accident theory suggests that a coherent twin boundary forms at a migrating grain boundary due to a stacking error [28]. The migration distance and migration velocity of the grain boundary are two key positive factors to annealing twin generation [29]. However, because of the introduction of the GNSs, more GNS-Cu interfaces formed in the composite and could inhibit the migration of the Cu grain boundaries, which would have a negative effect on the formation of the annealing twins. In addition, these GNSs also had a pining effect for dislocation movement, which could contribute to a higher dislocation density inside the GNS/Cu composites. In other words, the lattice stress level was higher in the GNS/Cu composite, which also explains the deformation and fracture features of the GNS/Cu composites.

### 3.3. Mechanical Properties of Pure Cu and GNS/Cu Composites

Figure 4 shows the room-temperature engineering tensile strain–stress curve of the hot-rolled pure Cu and the GNS/Cu composite. It is clearly seen that the yield strength (YS) of the GNS/Cu composite improved significantly in comparison with the pure Cu (from 169 MPa to 250 MPa). However, there was a sharp decrease in the elongation of the GNS/Cu composite. Chen et al. studied the effects of graphene content on the microstructure and properties of copper [30]. In their study, the GNS/Cu composites with graphene volume fractions of 0%, 0.2%, 0.4%, 0.6%, 0.8%, 2.0%, and 4.0% were fabricated through a molecular-level mixing method followed by an SPS process. Similar results were also observed in their work where the strengthening effects of GNSs deteriorated by increasing the graphene content to more than 2.0 vol.%. They attributed the deterioration in mechanical performance of the GNS/Cu composites to the weakened bonding and load transfer effect between the matrix and graphene. In this work, as the GNSs and the Cu particles were deposited alternatively, the volume fraction of the GNSs was calculated to be around 1.0 vol.% according to the thickness ratio between the GNS (~10 nm) and the Cu layer (~1 μm). Thus, the volume fraction of GNSs was comparable except that the fabrication method was different. The underlying fracture mechanism of the GNS/Cu composite is discussed in the next part.

### 3.4. Fracture Surface Morphologies of Pure Cu and GNS/Cu Composite

To determine the early failure of the GNS/Cu composites, fracture morphologies of the pure Cu and the GNS/Cu composite were characterized. Figure 5 shows the fractured surface morphologies of the hot-rolled pure Cu (Figure 5a,b) and the GNS/Cu composite (Figure 5c,d). It can be seen apparently that the pure Cu demonstrated a dimple pattern, which is a typical plastic fractured surface. For the GNS/Cu composite, plenty of GNSs were found peeled off from the Cu matrix and left in the bottom of the dimples (as pointed by the yellow arrows in Figure 5d), which was caused by the expansion of cracks that originated at the GNS-Cu interface. In addition, no stacking or agglomeration of GNSs was observed on the fractured surface, which indirectly evidenced the advantage of the EPD process used in this work for fabricating the GNS/Cu composite with a uniform GNS distribution. Notably, the Cu matrix stayed continuously in the GNS/Cu composite, although they were separated into thin layers by the incorporated GNSs. Inside the Cu layer, there still existed many dimples (as circled in Figure 5c); by contrast, the size of these dimples was much smaller compared with the dimples in the pure Cu, which indicates the toughness deterioration of the material. This may be because the incorporated GNSs split the Cu matrix into many thin layers at the micro-scale, and inside each Cu layer, there was not enough Cu volume to fulfil the plasticity. In addition, due to the poor affinity between Cu and GNSs, the micro-crack always initiated at the GNS-Cu interface. When increasing the applied load, these micro-cracks could expand and grow into large crack dimples with GNSs left at the bottom.

## 4. Discussion

To determine why micro-cracks always initiated at the GNS-Cu interface, local misorientations were analyzed in the GNS/Cu composite. As shown in Figure 6a–d, SEM–SE micrographs and the Kernel Average Misorientation (KAM) maps of the transverse area, which have different distances from the fracture surface, were constructed for the GNS/Cu composite. As displayed in Figure 6a,b, compared with the area that is 3 mm from the fracture (Figure 6a), the transverse area at the fracture (Figure 6b) had more and larger micro-cracks. Similarly, all the cracks were located at the positions with GNS distribution. As shown in Figure 6c,d, a higher level of misorientation, which inferred the strain level, was seen in the brighter green color in the grains around the crack tip with a severe deformation. It can be seen that the area around the GNSs always had a higher local misorientation compared with the area without GNSs. As the plastic deformation of the GNS/Cu composite in this work corresponds to dislocation slip, a higher local misorientation around the GNSs indicates a higher dislocation density and, therefore, higher stress concentration around the GNSs in the composite. This explains why the micro-cracks always initiated at the GNS-Cu interfaces. In addition, the effects of the soft Cu layer to relieve the stress concentration could be observed by analyzing the disorientation distribution in GNS/Cu composites with different Cu layers. Figure 6e,f display the disorientation from the average grain orientation spread (GOS) maps of 10% tensioned GNS/Cu samples with distinct Cu layer thicknesses (50 μm and 30 μm). We can clearly see that the area close to the GNS layer always possessed a higher disorientation, i.e., higher accumulated strain. Moreover, although the two kinds of samples possessed the same strain (10%), the GNS/Cu composite with the 30 μm thick Cu layer was in the higher strain state. This also proves that the soft Cu layer could help in relieving the accumulated stress and prevent the initiation of micro-cracks in the laminated GNS/Cu composites.

In order to prevent early failure of the GNS/Cu composites, enough ductile Cu matrix is a prerequisite to release excessive stress originated at the GNS-Cu interfaces. Figure 7 displays the room-temperature tensile curves of pure Cu and GNS/Cu composites with different Cu layer thicknesses. We found that the yield strength of all the GNS/Cu composites increased significantly compared with the pure Cu, especially the GNS/Cu composites with a thin Cu layer (≤10 μm). This is mainly contributed by the refined grain size as well as the increased back stress hardening induced by the incorporated GNSs. Owing to the incompatible plastic deformation behavior between the GNSs and the matrix Cu, deformation zones could form around the GNSs during the plastic deformation process, because the nuclei prefer to nucleate in the highly strained regions, i.e., the deformation zones [31]. In this way, the GNSs stimulated the nucleation by offering more nucleation sites, which belongs to the well-known mechanism named particle-stimulated nucleation (PSN) [32]. Moreover, the incorporated GNS layer could increase the dislocation density at the GNS-Cu interfaces by inhibiting the dislocation movements through the GNS layer. According to the back stress hardening mechanism [33], the accumulated dislocations could create a kind of long-range back stress to the other dislocations to make it difficult for them to slip anymore. It is observed that there was a great variation in the mechanical properties of the GNS/Cu composites with different Cu layer thicknesses. However, for the GNS/Cu composite with a Cu layer thickness of 10 μm, the mechanical performance was quite similar with the GNS/Cu composite fabricated by the alternative depositing method (Cu layer thickness less than 5 μm), and both demonstrated a brittle characteristic and fractured at the early tensile stage. Thus, we can see that the fracture behavior of the laminated GNS/Cu composites was strongly Cu-layer-thickness-dependent, and early failure of the composite could occur when the Cu layer thickness was too thin.

To understand the fracture mechanism of the GNS/Cu composites with different Cu layer thicknesses, as shown in Figure 8, fracture morphologies of GNS/Cu composites with Cu layer thicknesses of 10 μm, 30 μm, and 50 μm were characterized. For GNS/Cu composites with Cu layer thicknesses of 30 μm and 50 μm (Figure 8b,c), the layered structure was clearly observed. The dimples were located mainly at the GNS-Cu interface. Notably, these dimples were separated in different layers by the Cu layer, and a dimple connection was not found between neighboring layers. In this way, the plastic deformation occurred in different Cu layers. This may be the reason why these GNS/Cu composites showed fine ductility under applied loads. By contrast, for the GNS/Cu composites with a Cu layer thickness of 10 μm, they exhibited a similar fracture surface (as shown in Figure 8a) with the GNS/Cu composites prepared by the alternative depositing technique. As shown by yellow circles, connected dimples from different Cu layers were found, which could become micro-cracks in the composites. Because the Cu layer was too thin, the expanded dimples connected with the neighboring dimples easier, which led to the fracture of a single Cu layer in the GNS/Cu composite. Finally, the sample could fail when these kinds of cracks went through the whole sample.

Thus, the fracture mode of the GNS/Cu composites in this work can be illustrated by the diagram shown in Figure 9. As demonstrated in Figure 9a, before loading, the stress level was low in the composites and a layered structure was formed by the added GNSs. When the load was applied, as shown in Figure 9b, at the initial plastic deformation stage, dislocations were originated from the GNS-Cu interfaces, the homogeneously dispersed GNSs could increase the dislocation storage capacity of the composites so that the dislocation density of the composites enhanced intensively, and the area where GNSs were located reached a high stress level. This also explains why the GNS/Cu composites presented a higher hardening rate at the beginning of the tensile test. As the stress continued to increase (as shown in Figure 9c), the GNS-Cu interfaces that possessed the lowest strength in the composites started to debond and produce micro-cracks in the composites. As the loading continued, the micro-cracks in the composites tended to expand and connect with the neighboring cracks (shown in Figure 9d) and finally lead to the failure of the composites.

## 5. Conclusions

In this work, we present a novel alternating EPD technique to prepare GNS/Cu composite powder. Through this method, the GNSs realized a uniform distribution in the composite powder and the Cu-GNS-Cu layered structure was constructed in the GNS/Cu composite. The fracture behavior of the GNS/Cu composites with different Cu layer thicknesses was investigated. Specifically, the premature failure of the GNS/Cu with a thin Cu layer was thoroughly studied. The main conclusions are drawn as follows.

The alternating electrodeposition technique could be used to fabricate laminated GNS/Cu composites and the thickness of the Cu layer (equal to the diameter of the deposited Cu particles) in the Cu-GNS-Cu structure could be adjusted by altering the deposition duration.The yield strength of the GNS/Cu composites increased significantly compared with the Cu matrix due to grain refinement and the enhanced work hardening rate caused by the GNS addition. However, the mechanical properties (especially fracture behavior) of the laminated GNS/Cu composites were strongly determined by the Cu layer thickness, and early failure of the composite could occur when the Cu layer was too thin.The GNS/Cu composite combined the best strength and ductility when the Cu layer thickness was 30 μm. Early fracture may occur when the Cu layer was too thin (≤10 μm), while the strengthening efficiency decreased when the Cu layer thickness increased to 50 μm.The initiation of the microcracks in the GNS/Cu composites was caused by the excess stress concentration originated from the GNS/Cu interface. Sufficient Cu matrix could delay the microcracks from going through the Cu layer, which plays a key role in improving the ductility of the GNS/Cu composites.

## Figures and Tables

**Figure 1 materials-15-07677-f001:**
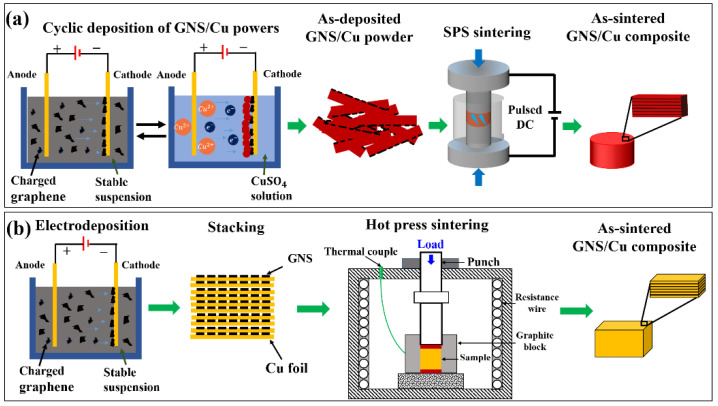
Schematic diagram for the fabrication of (**a**) laminated GNS/Cu composites by alternative depositing followed by SPS sintering and (**b**) laminated GNS/Cu composites by direct depositing of GNSs on cold-rolled Cu foils followed by hot-press sintering.

**Figure 2 materials-15-07677-f002:**
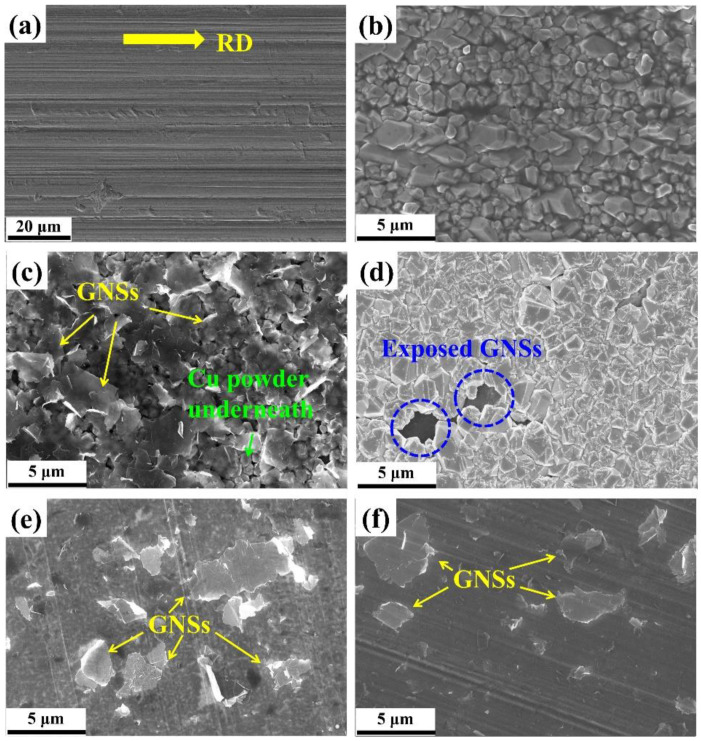
SEM–SE micrographs of (**a**) raw Cu foil, (**b**) Cu particles deposited on raw Cu foil, (**c**) GNSs deposited on Cu particles, (**d**) Cu particles deposited on GNSs, (**e**) 30 μm thick Cu foils deposited with GNSs, and (**f**) 50 μm thick Cu foils deposited with GNSs.

**Figure 3 materials-15-07677-f003:**
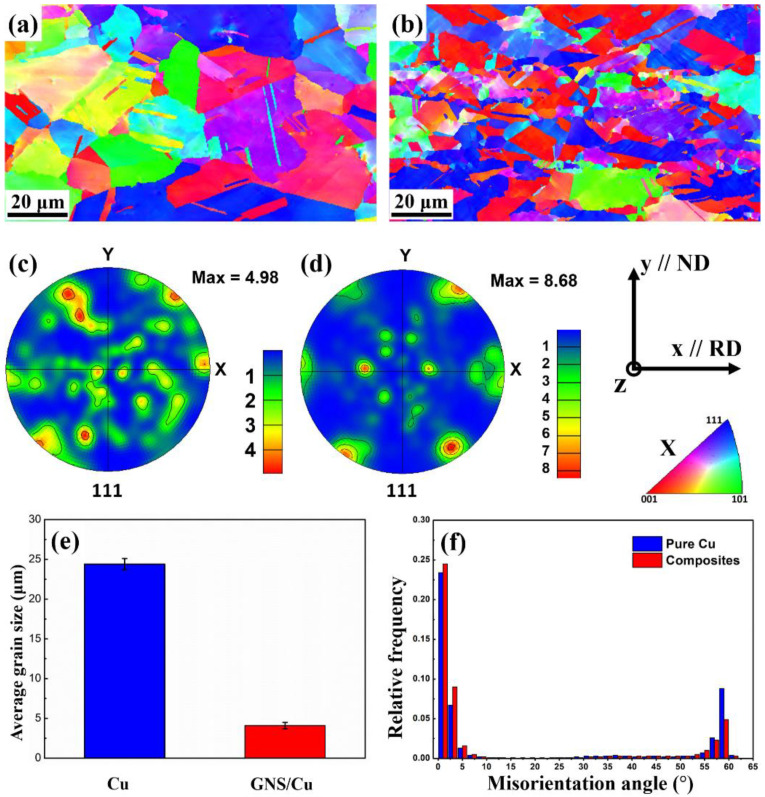
IPF-X images and {111} pole figures of hot-rolled (**a**,**c**) pure Cu and (**b**,**d**) GNS/Cu composites, and (**e**) average grain size and (**f**) misorientation angle distribution of hot-rolled pure Cu and GNS/Cu composites.

**Figure 4 materials-15-07677-f004:**
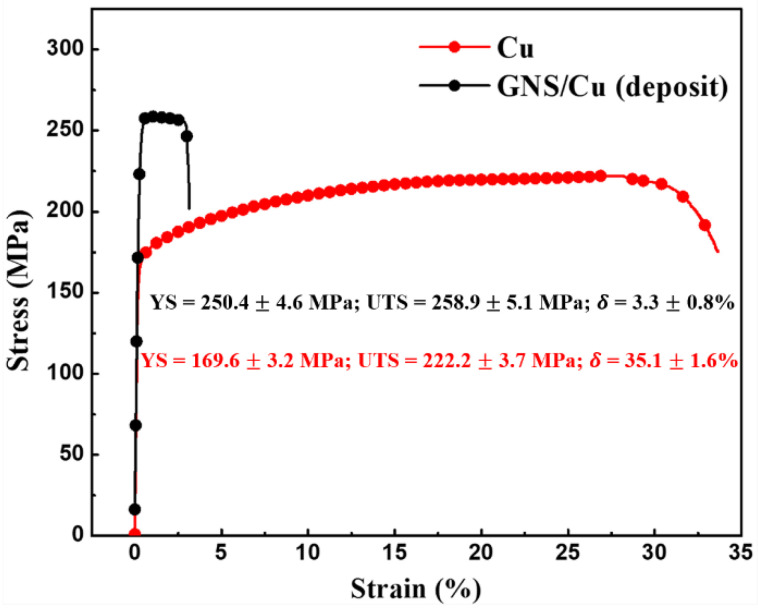
Room-temperature engineering strain–stress curve of pure Cu and GNS/Cu (deposit) composite.

**Figure 5 materials-15-07677-f005:**
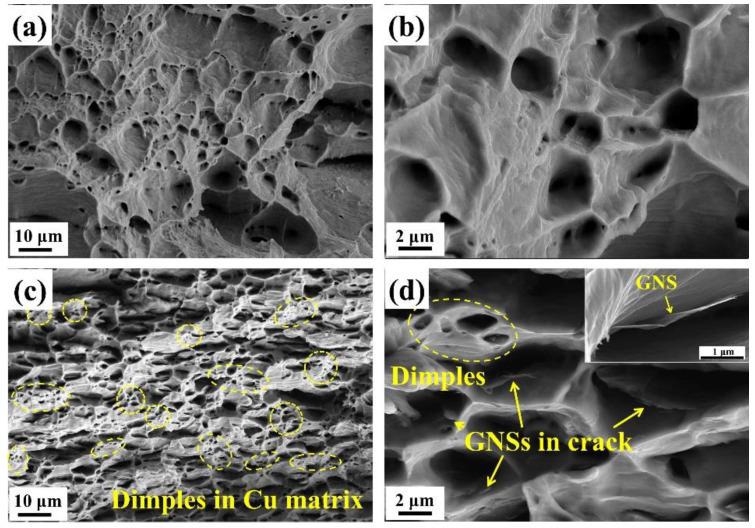
Fracture surface of (**a**,**b**) pure Cu and (**c**,**d**) GNS/Cu laminated composite (the yellow arrows indicate GNS inside the dimples).

**Figure 6 materials-15-07677-f006:**
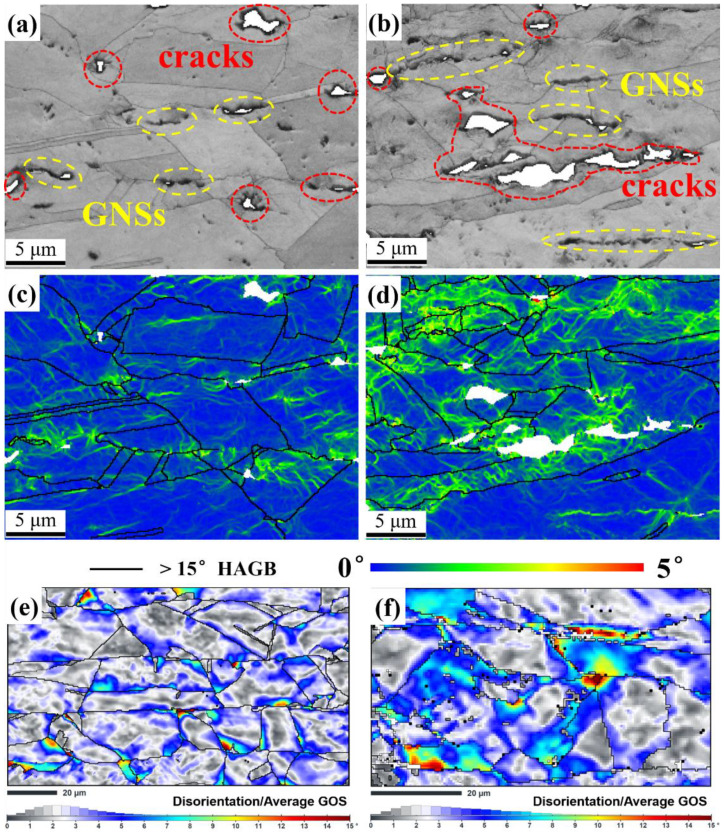
SEM–SE micrographs, Kernel Average Misorientation (KAM) maps, and disorientation from average grain orientation spread (GOS) maps of transverse area (**a**,**c**) 3 mm away from the fracture, (**b**,**d**) at the fracture, (**e**) 10% tensioned GNS/Cu (50 μm), and (**f**) 10% tensioned GNS/Cu (30 μm).

**Figure 7 materials-15-07677-f007:**
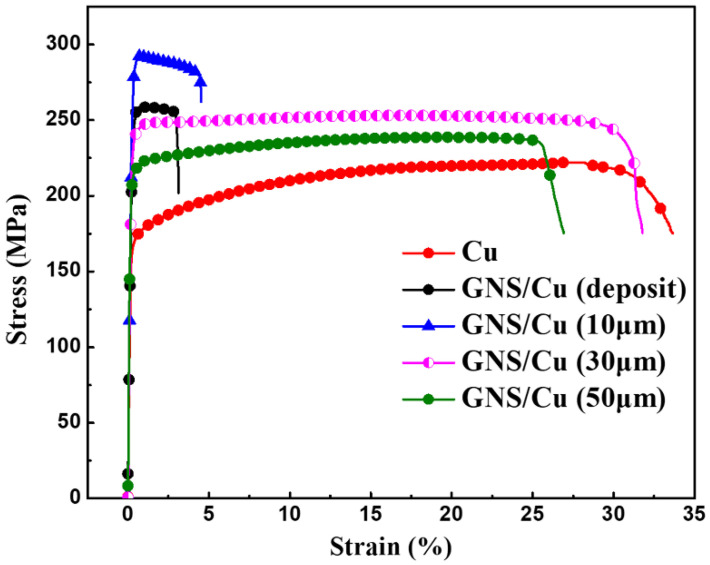
Room-temperature engineering strain–stress curves of pure Cu and GNS/Cu composites with various Cu layer thicknesses.

**Figure 8 materials-15-07677-f008:**
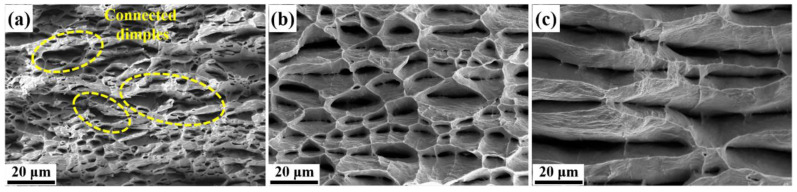
Fracture morphologies of GNS/Cu composites with initial Cu foil thickness of (**a**) 10 μm, (**b**) 30 μm, and (**c**) 50 μm.

**Figure 9 materials-15-07677-f009:**
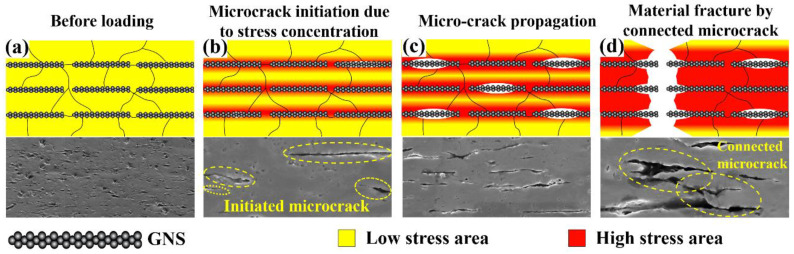
Schematic diagram of fracture mode of the GNS/Cu composites. (**a**) GNS/Cu composite before loading, (**b**) initiation of microcrack around the GNSs under applied load, (**c**) migration of microcrack in the GNS/Cu composite, and (**d**) connection of microcracks and fracture of the sample.

**Table 1 materials-15-07677-t001:** Chemical composition of the cold-rolled Cu foils (wt.%).

Cu	Pb	Fe	Sb	S	As	Bi
≥99.90	≤0.005	≤0.005	≤0.002	≤0.005	≤0.002	≤0.001

## Data Availability

The data presented in this study are available on request from the corresponding author.

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
