# Peer review of "Thin-Copper-Layer-Induced Early Fracture in Graphene-Nanosheets (GNSs)-Reinforced Copper-Matrix-Laminated Composites"

_materials, 2022, doi:10.3390/ma15217677_

Round 1
Reviewer 1 Report
The manuscript by Hailong Shi et a. describes the copper layer-induced early fracture in graphene nanosheets. The authors employed different Cu layer thicknesses produced by alternating electrodeposition technique followed by spark plasma sintering (SPS) and direct electrodeposition followed by hot press sintering. They show that the mechanical strength of the obtained Gr nanosheet/Cu campsite highly depends on the copper layer. Although the the questions addressed and result obtained are interesting, I have the following issues, which must be addressed before my final decision.
First of all, the authors published a very similar work two years ago (https://doi.org/10.1016/j.actamat.2020.09.017). I have to ask the authors t provide similarities and differences between the published and this work. Also, please specifically note the improvements from the past work.
My other concern is the extremely limited introduction, which covers only small aspects of the Gr and Gr-based composite/heterostructure activities. However, I believe that introducing more aspects of the research topic wi read more visibility and direct comparison of the employed methodologies. In this regard, recent activities comprise mainly the formation of Gr heterostructures/composites using intercalation of different species (such as: https://doi.org/10.1016/j.surfin.2022.102304 https://doi.org/10.1002/adfm.202109839 ), which is highly reliable and application perspective methods. I suggest the authors include such discussions also in the introductory part.
The authors claim the improved yield strength of the GNS/Cu composites compared with the Cu matrix because of the grain refinement. Although the authors relate grain refinement to the existence of the GNS, the reason why 2D GNS provides such new nucleation sites is missing from the discussions. I hope the authors can elaborate on this.
Although the authors employed two methods for composite preparation, their comparative discussion has not been provided. As a reader, this is one of the key aspects of this paper, and I hope the authors can extend discussions on this.
In addition to these specific questions, I recommend the authors to correct the existing errors and typos. Overall, I think the authors should resubmit the manuscript after addressing the questions/problems mentioned above.
Author Response
Dear editor,
We are very grateful to you and to the reviewers for the processing and the review of our manuscript entitled “Thin copper layer induced early fracture in graphene nanosheets (GNSs) reinforced copper matrix laminated composites”, by Hailong Shi, Xiaojun Wang, Xuejian Li, Xiaoshi Hu, Weimin Gan, Chao Xu and Guochao Wang.
We have carefully read the comments of the reviewers and revised the manuscript accordingly. For easy localization of the revised parts, we highlighted them in yellow in relation to the comments of Reviewer 1, in green in relation to the comments of Reviewer 2, in light blue in relation to the comments of Reviewer 3 and in pink in relation to the common comments of the reviewers. Hereunder are our responses.
-------------------------------------------------------------------------------------------------------
Reviewers' comments:
Reviewer #1:
manuscript by Hailong Shi et a. describes the copper layer-induced early fracture in graphene nanosheets. The authors employed different Cu layer thicknesses produced by alternating electrodeposition technique followed by spark plasma sintering (SPS) and direct electrodeposition followed by hot press sintering. They show that the mechanical strength of the obtained Gr nanosheet/Cu campsite highly depends on the copper layer. Although the questions addressed and result obtained are interesting, I have the following issues, which must be addressed before my final decision.
First of all, the authors published a very similar work two years ago (https://doi.org/10.1016/j.actamat.2020.09.017). I have to ask the authors t provide similarities and differences between the published and this work. Also, please specifically note the improvements from the past work.
Response: In our past paper entitled “Elastic strain induced abnormal grain growth in graphene nanosheets (GNSs) reinforced copper (Cu) matrix laminated composites”, we focused on the recrystallization behavior of 2D dimensioned Cu foils with the existence of GNSs. We used different kinds of characterization techniques including in-situ synchrotron diffraction, ex-situ EBSD and neutron diffraction to investigate the strain states, microstructure and texture evolution of the GNS/Cu composites. We found that the incorporated GNSs created a hybrid compressive strain states in the composites and resulted in the abnormal growth of the Cube-oriented grains. Differently, in this work, we focus on the deformation behavior of GNS/Cu composites with various thickness of Cu layers. Of course, this significant difference is also originated from the GNS layer. Moreover, the samples in our past paper are in the as-sintered state, while the samples were hot-rolled in this work. Thus, the two papers are totally different in cases of sample state, microstructure, research topic and employed characterization methods.
My other concern is the extremely limited introduction, which covers only small aspects of the Gr and Gr-based composite/heterostructure activities. However, I believe that introducing more aspects of the research topic wi read more visibility and direct comparison of the employed methodologies. In this regard, recent activities comprise mainly the formation of Gr heterostructures/composites using intercalation of different species (such as: https://doi.org/10.1016/j.surfin.2022.102304 https://doi.org/10.1002/adfm.202109839 ), which is highly reliable and application perspective methods. I suggest the authors include such discussions also in the introductory part.
Response: We agree with the reviewer that it would benefit the manuscript to add more information on graphene and graphene-based materials in the introduction part. We have also cited more papers related to this topic, including the as-mentioned two papers, in the revised manuscript. (Page 3, line 9-11)
The authors claim the improved yield strength of the GNS/Cu composites compared with the Cu matrix because of the grain refinement. Although the authors relate grain refinement to the existence of the GNS, the reason why 2D GNS provides such new nucleation sites is missing from the discussions. I hope the authors can elaborate on this.
Response: We have added the following discussion in the revised manuscript. (Page 18, line 1-13)
We found that the yield strength of all the GNS/Cu composites increased significantly compared with the pure Cu, especially the GNS/Cu composites with thin Cu layer (10 μm). This is mainly contributed by the refined grain size as well as the increased back stress hardening induced by the incorporated GNSs. Owing to the incompatible plastic deformation behavior between the GNSs and the matrix Cu, deformation zones could form around the GNSs during the plastic deformation process. Because the nuclei prefer to nucleate in the highly strained regions, i.e., the deformation zones [34]. In this way, the GNSs stimulated the nucleation by offering more nucleation sites, which belongs to the well-known mechanism named particle-stimulated nucleation (PSN) [35]. Moreover, the incorporated GNS layer could increase the dislocation density at the GNS-Cu interfaces by inhibiting the dislocation movements through the GNS layer. According to the back stress hardening mechanism [36], the accumulated dislocations could create a kind of long-range back stress to the other dislocations to make it difficult for them to slip anymore.
Although the authors employed two methods for composite preparation, their comparative discussion has not been provided. As a reader, this is one of the key aspects of this paper, and I hope the authors can extend discussions on this.
Response: Actually, in this work, the two kinds of fabrication methods, i.e., the alternating deposition + SPS sintering and the EPD + hot-press sintering are aimed to prepare GNS/Cu composites with different Cu layer thickness. Because it is really hard to purchase or to fabricate cold-rolled Cu foils thinner than 10μm, we adopted the alternating deposition method to fabricate GNS/Cu powder, which could introduce the Cu-GNS-Cu layered structure with Cu layer thickness smaller than 5μm. Thus, we believe that the Cu layer thickness is the most important factor which determine the mechanical properties of the laminated GNS/Cu composites.
In addition to these specific questions, I recommend the authors to correct the existing errors and typos. Overall, I think the authors should resubmit the manuscript after addressing the questions/problems mentioned above.
Response: We have revised the manuscript carefully to remove grammar errors and unnatural expressions as possible as we can. We have highlighted the revised parts in pink in the revised manuscript.

Reviewer 2 Report
Please see review report

Author Response
Dear editor,
We are very grateful to you and to the reviewers for the processing and the review of our manuscript entitled “Thin copper layer induced early fracture in graphene nanosheets (GNSs) reinforced copper matrix laminated composites”, by Hailong Shi, Xiaojun Wang, Xuejian Li, Xiaoshi Hu, Weimin Gan, Chao Xu and Guochao Wang.
We have carefully read the comments of the reviewers and revised the manuscript accordingly. For easy localization of the revised parts, we highlighted them in yellow in relation to the comments of Reviewer 1, in green in relation to the comments of Reviewer 2, in light blue in relation to the comments of Reviewer 3 and in pink in relation to the common comments of the reviewers. Hereunder are our responses.
-------------------------------------------------------------------------------------------------------
Reviewer #2:
Review of “Thin copper layer induced early fracture in graphene nanosheets (GNSs) reinforced copper matrix laminated compo-sites” submitted by Hailong Shi et al.
The authors in the manuscript conducted a comprehensive study on the graphene nanosheets reinforced copper matrix laminated composites by electrophoretic deposition. It is a useful method for strength-ductility trade-off of reinforced copper composites. However, the reviewer has the following concerns which need to be addressed before the manuscript should be published.
PRIMARY CONCERNS:
- The schematic diagram of Fig.1a seems to be confusing. After electrophoretic deposition, as-deposited GNS/Cu powder was prepared. Why is the GNS/Cu composites alignment neat in Fig.1a?
Response: We are grateful to the reviewer for this reminder and have modified the illustration in Fig. 1a. (Page 7, line 16)
- The writing needs to be clearer such that the readers can easily follow. Some of the key statements and points are difficult to understand.For example, “This may because the GNSs split the Cu matrix into many thin layers, inside which the Cu has not enough volume to fulfil its plasticity.”“the micro-crack always originate at the GNS-Cu interface”
Response: In the revised manuscript, we have carefully revised the expression to make it easier for the readers to follow. We have highlighted the revised parts in pink in the revised manuscript.
- The details regarding hot pressing are missing.
Response: We are sorry for this missed information and have added the hot-rolling details in the experimental part. (Page 7, line 10-15)
- The results should be written in correct forms, for example, the error should be added in the results of the grain sizes and tensile tests (included YS, UTS, et al.)
Response: We really appreciate this suggestion. In the revised manuscript, we have added the error bars to the experimental results accordingly. (Fig.2, Fig.6)
- The labels in the picture should be clear and visible, for example, Figure 2 and Figure 6. The scale bar should be the whole size. (Figure 6)
Response: In the revised manuscript, we have increased the font size or changed the font color of the labels in the images so that the items are clearly indicated. (Seen in Fig.2-6 and Fig. 9)
OTHER CONCERNS:
- The authors mentioned: “The initiation of the microcracks in the GNS/Cu composites was caused by the excess stress concentration originated from the GNS/Cu interface. Sufficient Cu matrix could delay the microcracks from going through the Cu layer which plays a key role in improving the ductility of the GNS/Cu composites.”. The authors give some fracture mechanisms, but there is not enough evidence to support the fracture mechanisms.
Response: We have added some new data (Figure 6e,6f) and also the corresponding discussions (Page 16, line 11-19) in the revised manuscript to further support the proposed fracture mechanism in this work.
- In the manuscript, the microstructure of GNS/Cu should be further investigated. Also, the distribution of GNS in the Cu matrix after SPS and hot pressing should be investigated
Response: In Fig.2, we presented the SEM-SE images of raw Cu foils, the deposited Cu particles, the deposited GNSs on the Cu particles, etc. In the revised manuscript, we also added the SEM-SE images of GNSs deposited on the initial cold-rolled Cu foils (Fig. 2e-f). Thus, the readers can have a better understanding of the samples prepared using two methods. However, because the 2D features of the GNS, it is really difficult to observe the distribution of GNSs by SEM-EBSD or even TEM once the GNS/Cu composite powder is sintered into the bulk material. In this work, instead, the distribution of GNSs in the composites are presented directly by the fracture morphologies (Fig. 5,7) and indirectly by the SEM-SE image of the tensioned sample (TD plane) (Fig. 9).

Reviewer 3 Report
1- The authors have tried to improve the mechanical properties of copper metal laminate through combining graphene nanosheets into the matrix. The strength of this work is to address the trade off between strength and ductility.
2- Nonmetallic-metal composite is another important attempt that has been widely used to address the shortcomings in lack of several mechanical properties. The combination between these two classes of materials can result in superior properties.
3- Using two sintering techniques; Spark plasma sintering (SPS) and hot pressing (HP) is another novel accountability to further improve the strength of Cu/GNS laminates.
4- The presentation of Scanning electron microscopies (SEM) micrographs is significant to relate the strength (from stress-strain curve) with the porosity and orientation of GNS inside the Cu laminates.
5- The flow of schematic representation of two methods and then SEM & IPF images is well prepared.
6- English language used throughout the manuscript is satisfactory.
However, there are several shortcomings that should be considered:
1- The authors displayed figure 7 to show the effect of Cu layers on the strength -ductility relationship. Nevertheless, not a concrete conclusion was made regarding selection of the best Cu/GNS thickness layer and not enough discussion is reported to explain this important trade off between both as mentioned in the abstract.
2- The comparison between two different methods (presented in figure 1) is not addressed in the discussion section. It is not clear which methodology results in better trade off between strength and ductility which was mentioned at the start of the abstract section.
3-I found this manuscript published in the journal Acta Materialia https://doi.org/10.1016/j.actamat.2020.09.017 the methodology described there is quite similar to that in the submitted manuscript, if they are not the same!
4-It is preferred the authors declare which composition (10 micrometer thickness, 30 micrometer thickness or 50 micrometer thickness) are the best to combine the best ductility and sustainable strength. and, it is better to mention their aim was achieving the sustainable yield strength (in case of ductility) or obtaining ultimate tensile strength.
- checking the reference NO: 17 style.
Author Response
Dear editor,
We are very grateful to you and to the reviewers for the processing and the review of our manuscript entitled “Thin copper layer induced early fracture in graphene nanosheets (GNSs) reinforced copper matrix laminated composites”, by Hailong Shi, Xiaojun Wang, Xuejian Li, Xiaoshi Hu, Weimin Gan, Chao Xu and Guochao Wang.
We have carefully read the comments of the reviewers and revised the manuscript accordingly. For easy localization of the revised parts, we highlighted them in yellow in relation to the comments of Reviewer 1, in green in relation to the comments of Reviewer 2, in light blue in relation to the comments of Reviewer 3 and in pink in relation to the common comments of the reviewers. Hereunder are our responses.
-------------------------------------------------------------------------------------------------------
Reviewer 3.
1- The authors have tried to improve the mechanical properties of copper metal laminate through combining graphene nanosheets into the matrix. The strength of this work is to address the trade off between strength and ductility.
2- Nonmetallic-metal composite is another important attempt that has been widely used to address the shortcomings in lack of several mechanical properties. The combination between these two classes of materials can result in superior properties.
3- Using two sintering techniques; Spark plasma sintering (SPS) and hot pressing (HP) is another novel accountability to further improve the strength of Cu/GNS laminates.
4- The presentation of Scanning electron microscopies (SEM) micrographs is significant to relate the strength (from stress-strain curve) with the porosity and orientation of GNS inside the Cu laminates.
5- The flow of schematic representation of two methods and then SEM & IPF images is well prepared.
6- English language used throughout the manuscript is satisfactory.
However, there are several shortcomings that should be considered:
1- The authors displayed figure 7 to show the effect of Cu layers on the strength -ductility relationship. Nevertheless, not a concrete conclusion was made regarding selection of the best Cu/GNS thickness layer and not enough discussion is reported to explain this important trade off between both as mentioned in the abstract.
Response: We are grateful to reviewer for this suggestion.
In the revised manuscript, we have added more discussions on the influence of Cu layer thickness on the trade-off between strength and ductility of the GNS/Cu composites. (Page 16, line 11-19)
2- The comparison between two different methods (presented in figure 1) is not addressed in the discussion section. It is not clear which methodology results in better trade off between strength and ductility which was mentioned at the start of the abstract section.
Response: Actually, in this work, the two kinds of fabrication methods, i.e., the alternating deposition + SPS sintering and the EPD + hot-press sintering are aimed to prepare GNS/Cu composites with different Cu layer thickness. Because it is really hard to purchase or to fabricate cold-rolled Cu foils thinner than 10μm, we adopted the alternating deposition method to fabricate GNS/Cu powder, which could introduce the Cu-GNS-Cu layered structure with Cu layer thickness smaller than 5μm. Thus, we believe that the Cu layer thickness is the most important factor which determine the mechanical properties of the laminated GNS/Cu composites.
3-I found this manuscript published in the journal Acta Materialia https://doi.org/10.1016/j.actamat.2020.09.017 the methodology described there is quite similar to that in the submitted manuscript, if they are not the same!
Response: The fabrication method for the samples investigated in our past paper published in the journal ‘Acta Materialia’ is similar to the GNS/Cu samples (Cu layer thickness of 10μm, 30μm and 50μm) studied in this work. However, they are not the same, all the samples were proceeded with hot-rolling process to further densify the composites in order to obtain better mechanical performance. Differently, in our past paper, we focused on the distinct recrystallization behavior induced by the incorporated GNSs. Thus, the as-sintered samples and samples sintered to different temperatures were investigated.
4-It is preferred the authors declare which composition (10 micrometer thickness, 30 micrometer thickness or 50 micrometer thickness) are the best to combine the best ductility and sustainable strength. and, it is better to mention their aim was achieving the sustainable yield strength (in case of ductility) or obtaining ultimate tensile strength.
Response: We agree with reviewer that it is necessary to declare the importance of the Cu layer thickness on the comprehensive mechanical properties of the GNS/Cu composites. We have added a new item of conclusion in Page 22, line 2-4.
- checking the reference NO: 17 style.
Response: We really appreciate the reviewer’s careful review of our manuscript and have corrected the reference No.17 in the revised manuscript. (Page 25, line 6)

Round 2
Reviewer 1 Report
The authors have adequately answered the reviewer's comments. The paper has been improved a lot after the revision.
I found the following inconsistency that must be corrected before the final acceptance. The references are placed incorrectly in the following sentence of the introduction. "Normally, high-quality graphene could be produced by a few methods, including micromechanical exfoliation [9], liquid-phase exfoliation (LPE) [10], chemical vapor deposition (CVD) [11] and element intercalation [12,13], etc."
after "element intercalation [12,13], please cite current ref number 10 only instead of 12 and 13. Ref 10 is intercalation not liquid-phase exfoliation (LPE). I think Ref 11 of the current bibliography does not need to be cited since it is about graphene nanoribbons.
Author Response
We appreciate the referee's careful review which really help to improve the quality of our manuscript.
The reference has been replaced accordingly, including the number of other references in the whole manuscript.
Reviewer 2 Report
The authors addressed my concerns.
Author Response
We thank the referee again for the careful review of our manuscript.